# Pemetrexed Hinders Translation Inhibition upon Low Glucose in Non-Small Cell Lung Cancer Cells

**DOI:** 10.3390/metabo11040198

**Published:** 2021-03-26

**Authors:** Marie Piecyk, Mouna Triki, Pierre-Alexandre Laval, Helena Dragic, Laura Cussonneau, Joelle Fauvre, Cédric Duret, Nicolas Aznar, Toufic Renno, Serge N. Manié, Cédric Chaveroux, Carole Ferraro-Peyret

**Affiliations:** 1Cancer Research Centre of Lyon, Université Lyon, INSERM 1052, CNRS 5286, 69008 Lyon, France; marie.piecyk@chu-lyon.fr (M.P.); mouna.triki@lyon.unicancer.fr (M.T.); pierre-alexandre.laval@lyon.unicancer.fr (P.-A.L.); helena.dragic@lyon.unicancer.fr (H.D.); joelle.fauvre@inserm.fr (J.F.); cedric.duret@lyon.unicancer.fr (C.D.); nicolas.aznar@lyon.unicancer.fr (N.A.); toufic.renno@lyon.unicancer.fr (T.R.); 2Hospices Civils de Lyon, Biopathology of Tumours, CPE, GHE Hospital, 69500 Bron, France; 3INRAE, Unité de Nutrition Humaine, Université Clermont Auvergne, UMR1019, 63122 Clermont-Ferrand, France; laura.cussonneau@inrae.fr; 4Inserm U1242, Centre de Lutte Contre le Cancer Eugène Marquis, Université de Rennes, 35042 Rennes, France; s.manie@rennes.unicancer.fr

**Keywords:** protein synthesis, pemetrexed, glucose availability, ER stress signaling, NSCLC

## Abstract

Genetic alterations in non-small cell lung cancers (NSCLC) stimulate the generation of energy and biomass to promote tumor development. However, the efficacy of the translation process is finely regulated by stress sensors, themselves often controlled by nutrient availability and chemotoxic agents. Yet, the crosstalk between therapeutic treatment and glucose availability on cell mass generation remains understudied. Herein, we investigated the impact of pemetrexed (PEM) treatment, a first-line agent for NSCLC, on protein synthesis, depending on high or low glucose availability. PEM treatment drastically repressed cell mass and translation when glucose was abundant. Surprisingly, inhibition of protein synthesis caused by low glucose levels was partially dampened upon co-treatment with PEM. Moreover, PEM counteracted the elevation of the endoplasmic reticulum stress (ERS) signal produced upon low glucose availability, providing a molecular explanation for the differential impact of the drug on translation according to glucose levels. Collectively, these data indicate that the ERS constitutes a molecular crosstalk between microenvironmental stressors, contributing to translation reprogramming and proteostasis plasticity.

## 1. Introduction

Non-small cell lung cancers (NSCLC), as many solid tumors, are characterized by genetic alterations causing metabolic rewiring [1]. A well-known example of this is that enhancement of glucose uptake and aerobic glycolysis favors the generation of energy and biomass, subsequently promoting tumor development [1,2,3,4,5,6]. However, poorly vascularized regions and the exacerbated metabolism of tumor cells can disrupt the nutritional homeostasis within the tumoral microenvironment, leading to decreased concentrations of several nutrients. Accordingly, glucose availability is often lower in different types of solid tumors [7,8,9]. Cell adaptation and tumor progression will thus depend on the cellular capacity to orchestrate a molecular and metabolic program to cope with such metabolic stress. One major key in this process relies on the level of activation of the nutrient sensors and their downstream molecular pathways that will modulate the cell response. The unfolded protein response (UPR) is one of those pathways. In response to glucose deprivation, UPR transiently inhibits protein synthesis and induces the production of chaperone molecules in order to restore endoplasmic reticulum (ER) homeostasis and promote cell survival [10]. The failure of this rescue mechanism results in apoptotic cell death [11]. Three ER stress (ERS) transducers, controlling three distinct axes of the UPR, have been identified so far. Each branch is defined by a class of transmembrane ER-resident signaling components: IRE1 (inositol requiring enzyme 1), PERK (double-stranded RNA-activated protein kinase (PKR)–like ER kinase), and ATF6 (activating transcription factor 6) [12]. The activation of the PERK axis plays a pivotal role for pulmonary cell outcome upon metabolic stress, particularly the phosphorylation of its substrate, the translation initiation factor eIF2a (eukaryotic initiation factor 2a) [13,14,15]. Indeed, eIF2a phosphorylation provokes a cytoprotective repression of protein synthesis and cell cycle arrest [16,17]. Conversely, upon prolonged stress, recovery of protein synthesis mediated by dephosphorylation of eIF2a and mTORC1 (mechanistic target of rapamycin complex 1) activation triggers cell death [18,19].

In addition to intrinsic metabolic stress, NSCLC cells are subjected to chemotherapeutic cues. Interestingly, some of those are now described for modulating PERK signaling in non-limiting nutrient conditions. For instance, in lung cancer cells, cisplatin treatment induces a protective response mediated by ERS [20]. Therefore, eIF2a represents a common integrator of signals arising from microenvironmental and therapeutic stresses to control translation.

Pemetrexed (PEM) is one of the most-extensively prescribed antifolate chemotherapeutic drugs for maintenance therapy of patients with locally advanced or metastatic NSCLC [21,22]. This agent impedes the synthesis of the pool of folate and purine, which is crucial for the generation of nucleotides and cell replication. In addition, production impairment of the pool of purine by PEM leads to ATP depletion [23]. Although protein synthesis and biomass generation represent the most consuming energy pathway in the cell, the impact of PEM on translation remains unclear in nutrient rich conditions. Furthermore, chemical activation of the PERK-eIF2a-ATF4 axis protects tumor cells from antifolate treatment through the induction of carbon metabolism [24]. However, whether protein synthesis may be differently modulated by PEM when eIF2a is phosphorylated remains unknown.

In this study, we thus determined the consequences of PEM on biomass and translation in glucose-abundant or scarce conditions. We show that NSCLC cells, treated with PEM, are prone to a de-repression of the rate of protein synthesis under low glucose availability. Our molecular investigations revealed that this effect is associated with a down-regulation of the UPR level.

## 2. Results

### 2.1. Pemetrexed Partially Reverts Low Glucose Inhibition of Protein Synthesis

To evaluate the effect of PEM on protein synthesis when glucose availability is low, we first set up experimental conditions reflecting glucose availability in tumors in vivo. To this end, A549 NSCLC cells were cultivated in medium containing 10 mM or 1 mM of glucose. These concentrations are proportional to those measured respectively in the plasma and tumor interstitial fluids of mice bearing xenografts [9]. Cell mass monitoring revealed an impairment by low glucose at 72 h (Figure 1a). To avoid data misinterpretation between loss of viability and biomass, we performed our experiments at 72 h, prior to proliferation arrest or standard cell death. At this specific time point, cell counting revealed that the number of viable A549 cells in 1 mM of glucose was similar to that in the 10 mM condition, irrespective of the treatment administered (Figure 1b,c). Western blot analysis confirmed that a 72 h-glucose starvation did not lead to caspase 3 and PARP cleavage (Figure 1d) [25,26]. This result is in accordance with the previously published data indicating that NSCLC cells are resistant to glucose metabolism impairment [27]. However, upon PEM, the apoptotic activation process was lower in starved cells compared to the normal condition, suggesting a protective role for glucose deprivation at later time points. 

Then cells were treated with PEM for 72 h, concomitantly to glucose starvation, and total protein mass was measured in each condition (Figure 2a). As previously observed, we found a significant loss of biomass when A549 cells were cultured in 1 mM of glucose compared to their counterparts cultivated in 10 mM. This demonstrates that low glucose alone reduces protein content by up to 43%. PEM treatment led to a 75% reduction in protein mass in the 10 mM glucose condition. However, in low glucose medium, PEM caused a lower decrease (50%) in protein mass compared to the vehicle, suggesting that this drug dysregulated the effect of low glucose on protein homeostasis.

To confirm that the PEM-mediated reduction in cell mass relied on an impaired translation, the rate of protein synthesis was assessed by using the SUnSET assay in the experimental conditions described above (Figure 2b). As expected, glucose deprivation provoked a dramatic (83%) decrease in puromycin incorporation. In non-limited glucose condition, PEM alone led to the repression (72%) of protein synthesis confirming that the reduction in biomass in response to the drug could be attributed to the repression of protein translation. However, when PEM-treated cells were maintained in 1 mM glucose, the rate of protein synthesis was significantly higher than that in the glucose-starved condition alone. These results show that PEM treatment partially relieves translation inhibition caused by low glucose.

Altogether, those data show that PEM differentially impacts the rate of protein synthesis according to the glucose availability.

### 2.2. Pemetrexed Treatment Alleviates UPR Induction in Low Glucose Condition

Next, we sought to determine the underlying mechanism by which PEM prevents low glucose-induced inhibition of protein synthesis. In glucose-starved cells, repression of protein synthesis is a consequence of various signaling pathways, including ERS signaling [28]. Particularly, activation of the PERK branch of the UPR leads to the inhibition of translation initiation and a selective elevation of the terminal transcription factor ATF4 (activation transcription factor 4). Thus, to investigate whether the UPR is affected in the different conditions, we first analyzed the degree of activation of ERS pathway at the studied time points (Figure 3a). We observed that ATF4 and ERS canonical target genes, CHOP (C/EBP homologous protein) and BiP (binding immunoglobin protein), increased after 48 h of glucose deprivation. Unlike CHOP, levels of ATF4 and BiP then decreased at 72 h, likely due to the action of molecular feedbacks on the UPR pathways [29,30], indicating that the peak of activation for ERS signaling occurs before 72 h of glucose starvation. Therefore, we then addressed whether induction of the UPR is impaired when glucose scarcity is associated with PEM at 48 h (Figure 3b). Glucose deprivation led to the activation of the ERS, as evidenced by a slower mobility of PERK, in addition to the accumulation of both ATF4 and CHOP and the canonical ATF6 marker BiP. PEM alone, compared to the vehicle, did not trigger ERS, as no change in these markers was observed. However, when glucose is low, PEM limits the increase in phosphorylated eIF2a, ATF4, CHOP, and BiP, further indicating that the weaker inducibility of the UPR by the agent is not restricted to the PERK axis. Expression analysis of canonical target genes of ATF4 (*ASNS* (asparagine synthetase), *TRB3* (tribble-3 related protein)) and ATF6 (*BIP*) confirmed the functional impairment of UPR signaling by PEM in deprived cells (Figure 3c). Indeed, although these three genes were upregulated upon low glucose, their inducibility was dramatically reduced when starvation was combined with PEM treatment.

Taken together, these results indicate that PEM limits UPR activation by low glucose levels, providing a mechanistic explanation as to the lower translation repression by the drug when glucose is scarce.

## 3. Discussion

This study aimed at determining whether PEM impairs protein homeostasis, and to unravel its crosstalk with glucose availability. Our data revealed that PEM treatment decreases translation activity in nutrient-rich conditions. However, in the context of repressed translation by glucose scarcity, PEM treatment dampened UPR activation and partially restored protein synthesis.

Our results provide a novel insight into the mode of action of PEM by showing that the drug counteracts protein biosynthesis when cells have a non-limited access to nutrients in a mechanism that is independent of the ERS signaling. In this context, the mode of action of PEM could implicate several molecular events converging to repressing protein synthesis. Knowing that PEM targets enzymes involved in decreasing nucleotide biosynthesis and pool, one could expect an alteration of mRNA synthesis [31,32,33,34]. However, this agent leads to an induction of PD-L1 expression in NSCLC cells, indicating that gene expression upon PEM is not totally repressed and might be selective [35]. As methotrexate, another antifolate compound, increases the AMP/ATP ratio, indicating that it depletes cells of ATP [23], PEM-induced reduction of the cellular pool of tetrahydrofolate likely diminishes the pool of ATP provoking a dysfunction of protein biosynthesis, one of the most consuming metabolic pathways [36]. Furthermore, alteration of energy metabolism represses the molecular axis that could contribute to translation inhibition. Indeed, several studies performed in complete medium reported that PEM inhibits mTORC1 signaling, a key regulator of cell growth, in an AMPK-dependent manner [37]. Repression of mTORC1 compromises the initiation and elongation stages through the dephosphorylation of 4EBP1 and S6K1/2 and by impairing ribosome biogenesis [38,39,40]. 

The nutritional context is emerging as a determinant of anticancer drug efficacy. Tumor cells evolving in nutrient-poor regions undergo dedifferentiation [41]. Acquisition of stemness features driven by metabolic stress including glucose starvation or hypoxia, triggers lower translation rates, particularly provoked by the repression of the mTOR pathway and phosphorylation of eIF2a [42,43]. Accordingly, rapamycin treatment, which mimics a starvation situation, antagonizes the cytotoxic effect in NSCLC cells [44,45]. The subsequent proliferation arrest and changes in tumor cell phenotype under stressed conditions might explain the resistance to PEM. Indeed, under severe hypoxia, expression of key enzymes and transporters in folate metabolism and nucleoside homeostasis is downregulated [46]. Raz et al. showed that hypoxia in solid tumors, by inducing cell cycle arrest, prevents antifolates from inducing DNA damage and apoptosis [47]. Furthermore, a recent work by Postovit’s group showed that the repression of mTOR caused by hypoxia or paclitaxel treatment promotes a 5′UTR selective translation of NANOG, SNAIL, and NODAL isoforms and the acquisition of stem-cell phenotypes and resistance to the drug [43]. Nonetheless, sensitivity to chemotherapy is restored upon ISRIB administration, a small molecule preventing eIF2a phosphorylation, by alleviating translation repression caused by stress [43]. 

Overall, further investigations are needed to comprehend whether the nutritional context impairs the efficacy of PEM treatment in clinics. In line with a previous study describing a protective role for the PERK pathway upon antimetabolites treatment [24], our results indicate that harsh nutritional microenvironment, reflecting the tumor glucose concentration, indeed protects against PEM in a lung cancer model mutated for KRAS. At the clinical level, these results propose the importance of assessing the stress degree in tumors for patient stratification and personalized adaptation of chemotherapeutic regimens. Recent data presented by the group of Dr. Koromilas confirm that increased phospho-eIF2a is associated with a poorer survival of patients [48]. Functionally, its pharmacological blockade by ISRIB prolonged survival of NSCLC mice models. These data, in conjunction with our findings, provide strongly compelling evidence to assess phospho-eIF2a in NSCLC tumors as a prognostic and stratification marker for PEM eligibility. Furthermore, the moderate translation recovery in starved cells upon PEM treatment attributed to partial induction of phospho-eIF2a axis likely participate in PEM resistance. Complete abolition of this stress integrator by ISRIB might represent a therapeutic vulnerability to ameliorate PEM sensitivity to NSCLC cells. 

## 4. Materials and Methods

### 4.1. Cell Culture and Treatments

A549 cells were purchased from the ATCC. They were routinely cultured in Dulbecco’s Modified Eagle Medium DMEM high glucose (GIBCO) supplemented with 10% fetal bovine serum (FBS) and 1% *v*/*v* penicillin/streptomycin (GIBCO) and were maintained at 37 °C in a 5% CO_2_ incubator. All of the treatments (chemotherapy and/or glucose deprivation) were performed the day after plating. Glucose deprivation was performed using Dulbecco’s Modified Eagle Medium devoid of glucose and sodium pyruvate (GIBCO). One percent penicillin/streptomycin (GIBCO), 1% sodium pyruvate, and 10% dialyzed FBS were subsequently added. The respective control medium for these experiments were also supplemented with glucose 200 g/L (GIBCO) to reach a final concentration of 10 or 1 mM. When necessary, chemotherapeutic treatment was performed daily using Pemetrexed generously provided by the Centre Léon Bérard at a final concentration of 4 µM. 

### 4.2. Cell Extracts and Western Blot Analysis

To perform Western blot analysis, A549 cells were seeded onto 6-well plates (2.50 × 10^5^ cells per well). Then, cells were treated with Pemetrexed and/or deprived of glucose for 72 h. Whole cell extracts were prepared from cultured cells lysed in RIPA protein buffer containing protease and phosphatase inhibitors (Roche) at 4 °C and obtained by centrifugation at 13,000× *g* for 20 min at 4 °C. Protein concentrations of the cellular extracts were determined using the DC Protein Assay (Bio-Rad, Hercules, CA, USA). Equal amounts of proteins (40 μg) were separated by SDS-PAGE and then transferred onto nitrocellulose membranes (Bio-Rad). Membranes were incubated in blocking buffer, 5% milk or Bovine Serum Albumin (BSA) in Tris-Buffered Saline/Tween 20 (TBST), for 1 h at room temperature, then incubated overnight at 4 °C with the appropriate primary antibodies, diluted in TBST containing 5% milk or BSA. Membranes were washed three times with TBST, incubated for 1 h at room temperature with the appropriate secondary antibodies, diluted in TBST containing 5% milk, and again washed three times with TBST. Detection by enhanced chemiluminescence was performed using the Clarity Western ECL substrate (Bio-Rad). Tubulin was used as a loading control. The primary antibodies used were purchased with the indicated dilution either from BD Biosciences: BiP (1/1000, 610978); Santa Cruz Biotechnology: ATF4 (1/1000, sc-200 and sc-390063), CHOP (1/200, sc-7351) and α-Tubulin (1/1000, sc-23950); from Cell Signaling Technology: PERK (1/1000, 3192S), c-Caspase 3 (1/1000, 9664), c-PARP (1/1000, 5625S), phospho-eIF2a (1/1000, 3398S); or from Millipore: Puromycin clone 12D10 (1/10,000, MABE343). The HRP-conjugated secondary antibodies were supplied by Cell Signaling Technology (1/10,000, anti-rabbit and anti-mouse antibodies, respectively, 7074S and 7076S). Western blot images are representative of three independent experiments. 

### 4.3. SUnSET Assay

Rates of protein synthesis were evaluated using the surface sensing of translation (SUnSET) method as previously described [49]. A549 cells were seeded onto 6-well plates (2.50 × 10^5^ cells per well) and treated with Pemetrexed and/or deprived of glucose for 72 h at 37 °C, 5% CO_2_. Fifteen minutes before being harvested and processed to prepare whole cell extracts in RIPA buffer, cells were incubated with 5 μg/mL of puromycin (Sigma P9620), directly added into the medium. The amount of puromycin incorporated into nascent peptides was then evaluated by Western blot analysis on 15 μg of proteins using anti-puromycin antibody purchased from Millipore (MABE343) and quantified with the ImageJ software (National Institutes of Health, Bethesda, MD, USA). 

### 4.4. RNA Extraction and RT-qPCR

Total cellular RNA was extracted after 48 h of treatment using TRIzol Reagent (Life Technologies) according to the manufacturer’s protocol. For cDNA synthesis, 0.5 µg of RNA were reverse transcribed using Superscript II reverse transcriptase (Invitrogen, ref: 18064014) with random primers (Invitrogen, ref: S0142), according to the manufacturer’s instructions. cDNA was then amplified by qPCR using specific primers listed below and the SYBR Green Master Mix (Bio-Rad). qPCR was performed using the CFX connect real-time PCR system (Bio-Rad). Relative quantification was assessed using a standard curve-based method. Expression of target genes (*TRB3* Fw: TGGTACCCAGCTCCTCTACG Rev: GACAAAGCGACACAGCTTGA; *BIP* Fw: CTGTCCAGGCTGGTGTGCTCT Rev: CTTGGTAGGCACCACTGTGTTC; *ASNS* Fw: CTGTGAAGAACAACCTCAGGATC; Rev: AACAGAGTGGCAGCAACCAAGC) was normalized against 3 endogenous mRNA levels (*18S* Fw: GAACGCCACTTGTCCCTCTA Rev: GTTGGTGGAGCGATTTGTCT, *Actin* Fw: TCCCTGGAGAAGAGCTACGA Rev: AGCACTGTGTTGGCGTACAG, *HPRT* Fw: TGACCTTGATTTATTTTGCATACC Rev: CGAGCAAGACGTTCAGTCCT), used as internal controls. QPCR experiments were repeated at least three times in duplicate. 

### 4.5. Cell Mass Assay

Cell mass biosynthesis was assessed by using the sulforhodamine B (SRB) assay. A549 cells were seeded onto 6-well plates (2.50 × 10^5^ cells per well) and treated with Pemetrexed and/or deprived of glucose for 24 to 72 h at 37 °C, 5% CO_2_. After removing the medium, wells were washed with PBS. Then, 1 mL of 10% trichloroacetic acid was added and after 1 h of incubation at 4 °C, the plates were flicked and washed three times with tap water before being dried at 37 °C for 1 h. The wells were then stained with 1.5 mL of SRB solution (0.057% in 1% acetic acid) for 30 min at room temperature under gentle agitation. Plates were flicked and washed three times with 1% acetic acid and air-dried for at least 1 h. Finally, 1.5 mL of 10mM Tris base was added and shaken vigorously for 15 min. The absorbance was measured using Tecan Infinite M200 Pro at a wavelength of 510 nm. 

### 4.6. Cell Viability Assay

Cell viability was measured using the trypan blue dye exclusion assay. A549 cells were seeded onto 6-well plates (2.5 × 10^5^ cells per well) and treated with Pemetrexed and/or deprived of glucose for 72 h at 37 °C, 5% CO_2_. First, the medium was collected and centrifuged twice at 500× *g* for 5 min. Dead cells were then resuspended in 100 µL and counted. Second, adherent cells were washed with PBS and detached with 500 µL of accutase (ref 11-007, GE Life Sciences/PAA Laboratories) for 10–15 min at 37 °C. Detached cells were resuspended in trypan blue dye at a ratio of 1:1. After 5 min of incubation, viable (stainless) and non-viable (blue) cells were counted in all studied conditions using a Glasstic slide from Kova International (ref: 87144).

### 4.7. Statistical Analyses

Statistical analyses were performed using the SigmaPlot software (Systat Software, San Jose, CA, USA) via one-way ANOVA with Tukey’s or Kruskal–Wallis multiple comparisons test. All data are expressed as means ± SEM of the indicated number of experiments, * *p* < 0.05, ** *p* < 0.01, *** *p* < 0.001.

## Figures and Tables

**Figure 1 metabolites-11-00198-f001:**
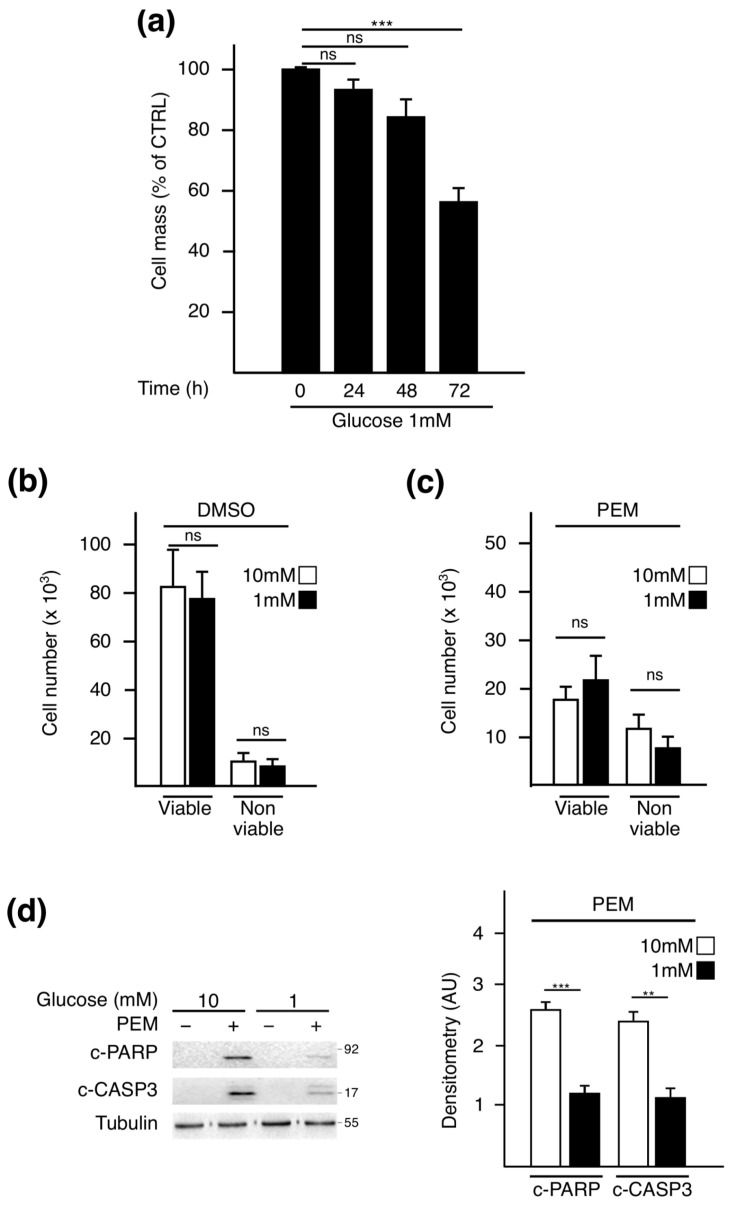
Experimental settings for assessing the impact of pemetrexed treatment on biomass independently of cytotoxicity. (**a**) Time course analysis of cell mass monitored by sulforhodamine B (SRB) assay performed on A549 cells cultured in 1 mM for the indicated time points. Control (CTRL) is referred to as the time point 0 h of deprivation. Data represent the mean of four independent experiments ± SEM. Cell viability was assessed using trypan blue exclusion assay after 72 h of culture in medium containing 10 mM or 1 mM glucose and treated with (**b**) DMSO or (**c**) PEM (4 µM). (**d**) Western blot analysis of apoptotic markers: cleaved forms of caspase 3 (c-CASP3) and PARP (c-PARP). For Western blotting, the most representative result from three independent experiments is displayed. Quantitative analysis was performed by comparing signals obtained in PEM-treated conditions normalized to tubulin using ImageJ software. ** *p* < 0.01, *** *p* < 0.001, ns: not significant.

**Figure 2 metabolites-11-00198-f002:**
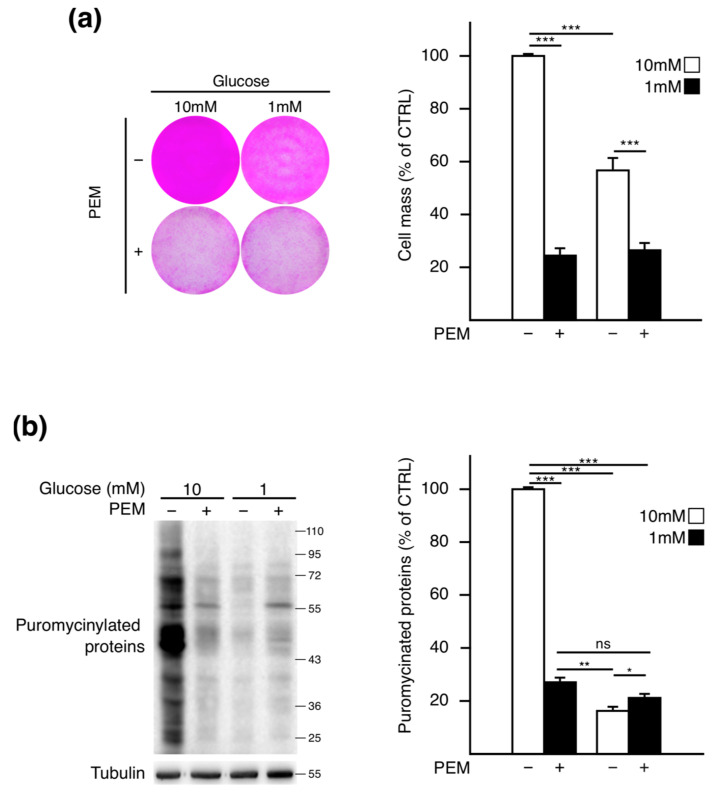
Protein synthesis is differently modulated according to glucose availability and pemetrexed treatment. (**a**) After 72 h, in 10 mM or 1 mM glucose-containing medium, A549 cells were treated or not with PEM (4 µM), and protein mass was measured using a SRB assay. SRB staining was imaged before measuring the corresponding absorbance. Control (CTRL) is referred to as the 10 mM without PEM condition. Data represent the mean ± SEM (n = 8). (**b**) The protein synthesis rate was assessed by SUnSET assay and quantified by measuring the ratio of puromycinated proteins normalized against tubulin using the ImageJ software. Quantification data represent the mean ± SEM (n = 3). * *p* < 0.05, ** *p* < 0.01, *** *p* < 0.001, ns: not significant.

**Figure 3 metabolites-11-00198-f003:**
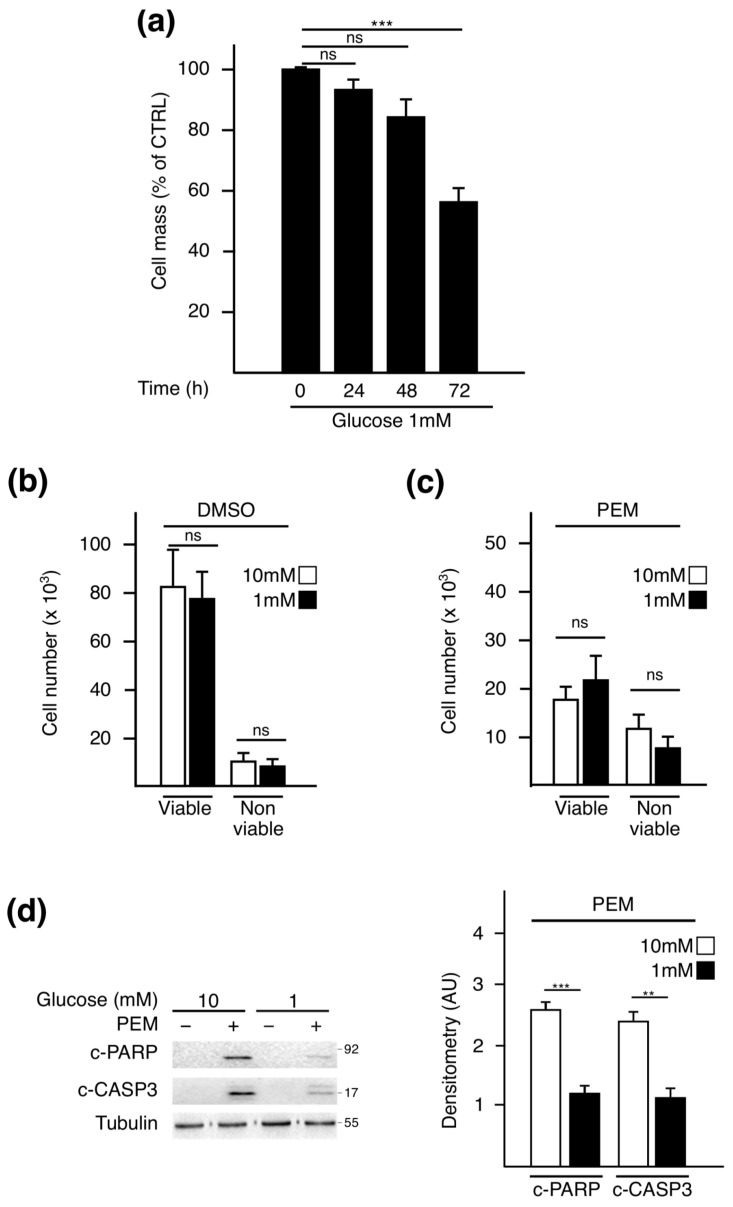
Pemetrexed constrains unfolded protein response (UPR) activation by glucose starvation. (**a**) Time course analysis of three ERS markers (ATF4, CHOP, BiP) at 48 h and 72 h following glucose deprivation. Quantification data represent the mean ± SEM (n = 3). (**b**) A549 cells were grown for 48 h in 10 mM or 1 mM glucose-containing medium combined or not with PEM (4 µM). Western blot analysis of UPR markers: PERK, phospho-eIF2a, ATF4, CHOP and BiP. O, inactivated and P, activated PERK. Quantification data represent the mean ± SEM (n = 3). For all Western blotting, the most representative result from three independent experiments is displayed, quantification of each marker was performed using the Image J software, and data are represented as fold change relative to the indicated condition. (**c**) Expression measurements of canonical UPR-target genes *ASNS*, *TRB3*, and *BIP*. QPCR data represent mean ± SEM (n = 3). * *p* < 0.05, ** *p* < 0.01, *** *p* < 0.001, ns: not significant.

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
