# Peer review of "Pemetrexed Hinders Translation Inhibition upon Low Glucose in Non-Small Cell Lung Cancer Cells"

_metabolites, 2021, doi:10.3390/metabo11040198_

Round 1

Reviewer 1 Report

The manuscript “Pemetrexed hinders translation inhibition upon low glucose in non-small cell lung cancer cells” by Marie et al. proposes a mechanism by which starved human lung adenocarcinoma cells respond when treated with pemetrexed (PEM). Data points to endoplasmic reticulum stress (ERS) signaling as a link between microenvironment and cell translation status. This study is of potential interest; however, some issues need further clarification.

The manuscript is well organized and well written. Nonetheless, it needs careful proofreading since the text has a few typos.

In the results section

  • In figure 2A: the graph represents cell mass as a percentage of the control. What was used as a control? Please include that information in the figure legend.
  • In figure 2B: the authors should include the molecular weight of the proteins analyzed by WB. The way that authors represent protein expression levels and the ratio is not reader-friendly. In this case, I would expect a graph with the quantitative analysis of each condition and corresponding error and statistical significance. How many independent experiments were conducted in this assay? The figure legend regarding panel 2B must be improved.
  • The authors mention that all treatments (chemotherapy and/or glucose deprivation) were performed the day after plating. However, the experiments lasted for different periods. The assay represented in Figures 1 and 2 were conducted after 72 hours, but the assay displayed in figure 3 was conducted after 48 hours. Why?
  • In figure 3A: the authors should include the molecular weight of the proteins that were analyzed by WB. This result would benefit if accompanied by quantitative analysis. How many independent experiments were conducted in this assay? On the right side of the WB image, there are two signs that are not mentioned in the figure legend. What do they mean?
  • In lines 116-17, the authors state, “Glucose deprivation led to the activation of the ER stress, as evidenced by a slower mobility of PERK”. How do the authors arrive at this conclusion? Also, the authors state “the accumulation of … ATF6 marker BiP”. It is not clear to me that BiP expression is increased, so I suggest a quantitative analysis of the WB.
  • Wouldn’t have been relevant to assess the phosphorylation status of PERK and/or eukaryotic initiation factor 2α (eIF2α)?

In the discussion section

  • Maybe the authors could elaborate more about the impact of this study in the clinic setting. How do these results contribute to improve therapeutic treatments?

In the materials and methods section

  • Please include the dilutions of primary and secondary antibodies used in the WB.

Author Response

Dear Reviewer 1,

Dear Reviewer, I would like to thank you for your reviews and comments. We appreciate the very constructive and important suggestions that you provided. We now provide new data that strengthen the manuscript and we hope that this revised version will address your concerns. Our responses to each comment can be found below.

Point-by-point response to the reviewer comments

The manuscript is well organized and well written. Nonetheless, it needs careful proofreading since the text has a few typos.

We thank the reviewer for warning us. Consequently, the revised version has been reviewed and edited by the Dr. Brigitte Manship, the scientific writer of the Cancer Research Center of Lyon.

In the results section

In figure 2A: the graph represents cell mass as a percentage of the control. What was used as a control? Please include that information in the figure legend.

For clarification, we have now indicated in the figure legend of panel 2A that the 10mM without PEM condition represents the experiment control.

In figure 2B: the authors should include the molecular weight of the proteins analyzed by WB. The way that authors represent protein expression levels and the ratio is not reader-friendly. In this case, I would expect a graph with the quantitative analysis of each condition and corresponding error and statistical significance. How many independent experiments were conducted in this assay? The figure legend regarding panel 2B must be improved.

The molecular weights and number of experiments are now indicated for all western blots including panel 2B. This panel has been quantified and analyzed for statistical relevance. The figure legend has been improved by providing the number of experiments and the method used for data quantification.

The authors mention that all treatments (chemotherapy and/or glucose deprivation) were performed the day after plating. However, the experiments lasted for different periods. The assay represented in Figures 1 and 2 were conducted after 72 hours, but the assay displayed in figure 3 was conducted after 48 hours. Why?

We agree with the reviewer that these aspects required clarifications. To illustrate the consequence of glucose deprivation on cell mass, we now provide, in Figure 1a, a time course analysis of cell mass. The result shows that functional loss of biomass occurs at 72h, without inducing cell toxicity. For these reasons, the consequence on translation of the different applied stresses was studied at this specific time point.

At the molecular level, the same approach was done for the UPR markers. In figure 3a, we established that the accumulation of canonical markers of the ER stress signaling, including ATF4, BiP and CHOP, is observed at 48 hours and then start to decrease at 72h. These results are consistent with the classical transient activation of the ER stress branches whose activities are subjected to negative molecular feedbacks notably controlled by p58 or GADD34.

In figure 3A: the authors should include the molecular weight of the proteins that were analyzed by WB. This result would benefit if accompanied by quantitative analysis. How many independent experiments were conducted in this assay? On the right side of the WB image, there are two signs that are not mentioned in the figure legend. What do they mean?

As mentioned above, the molecular weights are now indicated for all western blotting experiments including now Figure 3b. Moreover, all analyzed markers have been quantified and analyzed for statistical relevance. The number of independent experiments is now provided for all western blotting. We apologize for having forgotten to indicate the meaning of the mentioned signs. This mistake has been corrected and their meanings, as previously published (PMID: 27255611, 23395000, 25986605) are now provide in the figure legend: O, inactivated and P, activated PERK.

In lines 116-17, the authors state, “Glucose deprivation led to the activation of the ER stress, as evidenced by a slower mobility of « PERK”. How do the authors arrive at this conclusion? Also, the authors state “the accumulation of … ATF6 marker BiP”. It is not clear to me that BiP expression is increased, so I suggest a quantitative analysis of the WB.

Wouldn’t have been relevant to assess the phosphorylation status of PERK and/or eukaryotic initiation factor 2α (eIF2α)?

We have now provided a quantification of all Western blot for clarification. Regarding the reviewer’s concern about BiP, this quantification shows that this marker is no more induced by glucose deprivation when combined with PEM (Figure 3B). This result is consistent with the expression data presented in Figure 3c.

Detection of the phosphorylated form of PERK is recognized for being “challenging, if not impossible” (see PMID 21266248). We tried to detect this phosphorylation by immunoblotting but the results were not convincing even in the glucose-starved condition or with a classical chemical stressor (thapsigargin). For this reason, the characteristic mobility shift for PERK and IRE1 as well, observable by SDS-PAGE experiment, is now accepted as reflecting the active form of the two ER sensors (PMID: 20661282, 9930704, 25986605). We do observe this mobility shift of PERK in glucose-starved cells which is consistent with the elevation of ATF4 and CHOP as canonical markers of the PERK branch of the ER stress. Coherently this shift is reduced once combined with Pemetrexed and is associated with a weaker detection of the same markers.

Regarding the phosphorylation of eIF2a, we now provide the corresponding western blot in Figure 3B. The elevation of phospho-eIF2a in glucose starved cells is not visible at this specific time. It is well established that the kinetic of eIF2a phosphorylation is transient and occurs in a very early phase of stress. This kinetic of phosphorylation does not match the kinetic of ATF4 and CHOP elevations, which appear later (PMID 24648524). In our study, we focused at i) 48h of starvation for functionally assessing the consequence of the PEM-mediated activation on the ATF4-target genes and this point is likely too late for visualizing the increased phospho-eIF2a by -Glc and ii) 72 h of starvation for functional consequence on biomass (Figure 1a).  Nonetheless, in 1mM of glucose, the P- eIF2a band in the PEM-treated cells is fainter, attested by the quantification, than the one in the vehicle treated counterpart. This result is consistent with the behavior of the ER stress markers (ATF4, CHOP, ASNS, BiP) at the mRNA and protein level.

In the discussion section

Maybe the authors could elaborate more about the impact of this study in the clinic setting. How do these results contribute to improve therapeutic treatments?

Considering the new data showing a weaker elevation of proapoptotic markers by pemetrexed when glucose is scarce, we have extended the discussion section to the clinical interest of our results (last paragraph lines 279-294). We suggested that i) Phospho-EiF2a should be studied as a stratification marker for PEM eligibility ii) ISRIB, a Phospho-EiF2a pharmacological inhibitor, should be evaluated in combination to PEM, as a nax strategy to overcome NSCLC resistance to PEM.

In the materials and methods section

Please include the dilutions of primary and secondary antibodies used in the WB.

The antibody dilutions used for the different western blotting are now indicated in the material and method section.

Reviewer 2 Report

In this contribution, the authors reported that, under low glucose conditions (1 mM), treatment with PEM could decrease the inhibition of protein synthesis in A549 cells, which is likely caused by ERS. These results are interesting, I support the publication of this work after the authors address the following issues:

In Figure 3, it seems like the expression of PERK is at a higher level in PEM-treated group (low glucose conditions) as compared to that in non-treated group. Also, the PERK expression level is much lower under low glucose conditions (as compared to the level under high glucose conditions). Have you tried to monitor the phosphorylation level of PERK after various treatments? What’s the expression level of CHOP under these conditions?

ERS stress is often linked with apoptosis. Is there any data to show such connection?

In Figure 3, the descriptions for b, c, and d do not match with labels in the Figure. Please check and make the correction.

Add reference(s) to, page 2, line 47-49, “The activation of the...”.

Check the references format. The author may want to use proper/consistent abbreviations for the names of the journals.

For the discussion part, the authors may want to focus more on the analysis derived from the results presented in this manuscript.

Author Response

Dear Reviewer 2,

Dear Reviewer, I would like to thank you for your reviews and comments. We appreciate the very constructive and important suggestions that you provided. We now provide new data that strengthen the manuscript and we hope that this revised version will address your concerns. Our responses to each comment can be found below.

Point-by-point response to the reviewer comments

In Figure 3, it seems like the expression of PERK is at a higher level in PEM-treated group (low glucose conditions) as compared to that in non-treated group. Also, the PERK expression level is much lower under low glucose conditions (as compared to the level under high glucose conditions). Have you tried to monitor the phosphorylation level of PERK after various treatments? What’s the expression level of CHOP under these conditions?

- Detection of the phosphorylated form of PERK is recognized for being “challenging, if not impossible” (see PMID 21266248). We tried to detect this phosphorylation by immunoblotting but the results were not convincing even in the glucose-starved condition or with a classical chemical stressor (thapsigargin). For this reason, the characteristic mobility shift for PERK and IRE1 as well, observable by SDS-PAGE experiment, is now accepted as reflecting the active form of the two ER sensors (PMID: 20661282, 9930704, 25986605). We do observe this mobility shift of PERK in glucose-starved cells which is consistent with the elevation of ATF4 and CHOP as canonical markers of the PERK branch of the ER stress. Coherently this shift is reduced once combined with Pemetrexed and is associated with a weaker detection of the same markers.

- Regarding the CHOP expression level in these condition, new data in Figure 3b revealed that glucose deprivation alone leads to the expectedly augmentation of CHOP amount. However, in PEM-treated cells, the inducibility of CHOP is weaker and behave similarly to what was observed for ATF4 and BiP by the glucose deprivation.

ERS stress is often linked with apoptosis. Is there any data to show such connection?

We do agree with the reviewer that the relationship between ER Stress, apoptosis and drug resistance would represent an important point of this work. However, the scope of this study was to determine whether the ER Stress could represent a molecular node for translation regulation between chemotherapeutic and nutritional stresses and all the experimental settings were adjusted to avoid results misunderstanding with cell toxicity, as explained in the first part of the result section.

Nonetheless, we addressed by western blot the status of apoptotic markers induced by the ER stress by analyzing the amount of cleaved-PARP and cleaved-Caspase 3 (cf new Figure 1d). These two markers are classically related to the mechanism of apoptosis downstream the ER stress signaling (PMID 19360473, 28841673). Consistently with the described resistance of lung adenocarcinoma cells towards glucose deprivation (PMID 28548087), no change for these markers was observed after a 72 hour-glucose starvation despite the clear ER stress activation. However, the induction of these factors upon pemetrexed treatment is reduced in low glucose suggesting a protective role for the starvation at later time points, likely through the PERK branch as previously shown in PMID:32522993.

In Figure 3, the descriptions for b, c, and d do not match with labels in the Figure. Please check and make the correction.

We thank the reviewer for this remark and these mistakes have been corrected.

Add reference(s) to, page 2, line 47-49, “The activation of the...”.

We agree with the reviewer that bibliography references were lacking for this part. We now support the corresponding sentence by adding the following references (PMID: 27629041, 23395000, 24605820).

Check the references format. The author may want to use proper/consistent abbreviations for the names of the journals.

The bibliography section has been now formatted with the most recent MDPI style available in the Zotero library.

For the discussion part, the authors may want to focus more on the analysis derived from the results presented in this manuscript.

Considering the new data showing a weaker elevation of proapoptotic markers by pemetrexed when glucose is scarce, we have extended the discussion section to the clinical interest of our results (last paragraph lines 279-295). We suggested that i) Phospho-EiF2a should be studied as a stratification marker for PEM eligibility ii) ISRIB, a Phospho-EiF2a pharmacological inhibitor, should be evaluated in combination to PEM, as a new strategy to overcome NSCLC resistance to PEM.

Round 2

Reviewer 1 Report

The manuscript “Pemetrexed hinders translation inhibition upon low glucose in non-small cell lung cancer cells” by Piecyk et al. is substantially improved. The authors have answer to all my concerns.

Reviewer 2 Report

The authors have addressed my previous concerns, I support the publication of this manuscript.